# Learning Parametric Closed-Loop Policies for Markov Potential Games

**Sergio Valcarcel Macua**
PROWLER.io
Cambridge, UK
`sergio@prowler.io`

**Javier Zazo,  Santiago Zazo**
Information Processing and Telecommunications Center
Universidad Politécnica de Madrid
Madrid, Spain
`javier.zazo.ruiz@upm.es`
`santiago@gaps.ssr.upm.es`

## Abstract

Multiagent systems where the agents interact among themselves and with an stochastic environment can be formalized as stochastic games. We study a subclass of these games, named Markov potential games (MPGs), that appear often in economic and engineering applications when the agents share some common resource. We consider MPGs with continuous state-action variables, coupled constraints and nonconvex rewards. Previous analysis followed a variational approach that is only valid for very simple cases (convex rewards, invertible dynamics, and no coupled constraints); or considered deterministic dynamics and provided open-loop (OL) analysis, studying strategies that consist in predefined action sequences, which are not optimal for stochastic environments. We present a closed-loop (CL) analysis for MPGs and consider parametric policies that depend on the current state and where agents adapt to stochastic transitions. We provide easily verifiable, sufficient and necessary conditions for a stochastic game to be an MPG, even for complex parametric functions (e.g., deep neural networks); and show that a closed-loop Nash equilibrium (NE) can be found (or at least approximated) by solving a related optimal control problem (OCP). This is useful since solving an OCP—which is a single-objective problem—is usually much simpler than solving the original set of coupled OCPs that form the game—which is a multiobjective control problem. This is a considerable improvement over the previously standard approach for the CL analysis of MPGs, which gives no approximate solution if no NE belongs to the chosen parametric family, and which is practical only for simple parametric forms. We illustrate the theoretical contributions with an example by applying our approach to a noncooperative communications engineering game. We then solve the game with a deep reinforcement learning algorithm that learns policies that closely approximates an exact variational NE of the game.

## 1 Introduction

In a noncooperative *stochastic dynamic game*, the agents compete in a time-varying environment, which is characterized by a discrete-time dynamical system equipped with a set of states and a state-transition probability distribution. Each agent has an instantaneous reward function, which can be stochastic and depends on agents' actions and current system state. We consider that both the state and action sets are subsets of real vector spaces and subject to coupled constraints, as usually required by engineering applications.

A dynamic game starts at some initial state. Then, the agents take some action and the game moves to another state and gives some reward values to the agents. This process is repeated at every time step over a (possibly) infinite time horizon. The aim of each agent is to find the policy that maximizes its expected long term return given other agents' policies. Thus, a game can be represented as a set of *coupled* optimal-control-problems (OCPs), which are difficult to solve in general.

OCPs are usually analyzed for two cases namely *open-loop* (OL) or *closed-loop* (CL), depending on the information that is available to the agents when making their decisions. In the OL analysis, the

action is a function of time, so that we find an optimal sequence of actions that will be executed in order, without feedback after any action. In the CL setting, the action is a mapping from the state, usually referred as *feedback policy* or simply *policy*, so the agent can adapt its actions based on feedback from the environment (the state transition) at every time step. For deterministic systems, both OL and CL solutions can be optimal and coincide in value. But for stochastic system, an OL strategy consisting in a precomputed sequence of actions cannot adapt to the stochastic dynamics so that it is unlikely to be optimal. Thus, CL are usually preferred over OL solutions.

For dynamic games, the situation is more involved than for OCPs, see, e.g., (Basar and Olsder, 1999). In an OL dynamic game, agents' actions are functions of time, so that an OL equilibrium can be visualized as a set of state-action *trajectories*. In a CL dynamic game, agents' actions depend on the current state variable, so that, at every time step, they have to consider how their opponents would react to deviations from the equilibrium trajectory that they have followed so far, i.e., a CL equilibrium might be visualized as a set of *trees* of state-action trajectories. The sets of OL and CL equilibria are generally different even for deterministic dynamic games (Kydland, 1975; Fudenberg and Levine, 1988).The CL analysis of dynamic games with continuous variables is challenging and has only be addressed for simple cases.

The situation is even more complicated when we consider *coupled constraints*, since each agent's actions must belong to a set that depends on the other agents' actions. These games, where the agents interact strategically not only with their rewards but also at the level of the feasible sets, are known as generalized Nash equilibrium problems (Facchinei and Kanzow, 2010).

There is a class of games, named *Markov potential games* (MPGs), for which the OL analysis shows that NE can be found by solving a single OCP; see (González-Sánchez and Hernández-Lerma, 2013; Zazo et al., 2016a) for recent surveys on MPGs. Thus, the benefit of MPGs is that solving a single OCP is generally simpler than solving a set of coupled OCPs. MPGs appear often in economics and engineering applications, where multiple agents share a common resource (a raw material, a communication link, a transportation link, an electrical transmission line) or limitations (a common limit on the total pollution in some area). Nevertheless, to our knowledge, none previous study has provided a practical method for finding CL Nash equilibrium (CL-NE) for continuous MPGs.

Indeed, to our knowledge, no previous work has proposed a practical method for finding or approximating CL-NE for *any* class of Markov games with continuous variables and coupled constraints. State-of-the-art works on learning CL-NE for general-sum Markov games did not consider coupled constraints and assumed finite state-action sets (Prasad et al., 2015; Pérolat et al., 2017).

In this work, we extend previous OL analysis due to Zazo et al. (2016b); Valcarcel Macua et al. (2016) and tackle the CL analysis of MPGs with coupled constraints. We assume that the agents' policies lie in a *parametric* set. This assumption makes derivations simpler, allowing us to prove that, under some potentiality conditions on the reward functions, a game is an MPG. We also show that, similar to the OL case, the Nash equilibrium (NE) for the approximate game can be found as an optimal policy of a related OCP. This is a practical approach for finding or at least approximating NE, since if the parametric family is expressive enough to represent the complexities of the problem under study, we can expect that the parametric solution will approximate an equilibrium of the original MPG well (under mild continuity assumptions, small deviations in the parametric policies should translate to small perturbations in the value functions). We remark that this parametric policy assumption has been widely used for learning the solution of single-agent OCPs with continuous state-action sets; see, e.g., (Konda and Tsitsiklis, 2003; Melo and Lopes, 2008; Powell and Ma, 2011; Van Hasselt, 2012; Lillicrap et al., 2015; Heess et al., 2015; Schulman et al., 2015). Here, we show that the same idea can be extended to MPGs in a principled manner.

Moreover, once we have formulated the related OCP, we can apply reinforcement learning techniques to find an optimal solution. Some recent works have applied *deep reinforcement learning* (DRL) to *cooperative* Markov games (Foerster et al., 2017; Sunehag et al., 2017), which are a particular case of MPGs. Our results show that similar approaches can be used for more general MPGs.

**Summary of contributions.** We provide sufficient and necessary conditions on the agents' reward function for a stochastic game to be an MPG. Then, we show that a closed-loop Nash equilibrium can be found (or at least approximated) by solving a related optimal control problem (OCP) that is similar to the MPG but with a single-objective reward function. We provide two ways to obtain the reward function of this OCP: *i)* computing the line integral of a vector field composed of the partial

derivatives of the agents' reward, which is theoretically appealing since it has the form of a potential function but difficult to obtain for complex parametric policies; *ii)* and as a separable term in the agents' reward function, which can be obtained easily by inspection for any arbitrary parametric policy. We illustrate the proposed approach by applying DRL to a noncooperative Markov game that models a communications engineering application (in addition, we illustrate the differences with the previous standard approach by solving a classic resource sharing game analytically in the appendix).

## 2 PROBLEM SETTING FOR CLOSED-LOOP MPG

Let $\mathcal{N} \triangleq \{1, \ldots, N\}$ denote the set of agents. Let $a_{k,i}$ be the real vector of length $A_k$ that represents the action taken by agent $k \in \mathcal{N}$ at time $i$, where $\mathbb{A}_k \subseteq \mathbb{R}^{A_k}$ is the set of actions of agent $k \in \mathcal{N}$. Let $\mathbb{A} \triangleq \prod_{k \in \mathcal{N}} \mathbb{A}_k$ denote the set of actions of all agents that is the Cartesian product of every agent's action space, such that $\mathbb{A} \subseteq \mathbb{R}^A$, where $A = \sum_{k \in \mathcal{N}} A_k$. The vector that contains the actions of all agents at time $i$ is denoted $a_i \in \mathbb{A}$. Let $\mathbb{X} \subseteq \mathbb{R}^S$ denote the set of states of the game, such that $x_i$ is a real vector of length $S$ that represents the state of the game at time $i$, with components $x_i(s)$:

$$x_i \triangleq (x_i(s))_{s=1}^S \in \mathbb{X}. \tag{1}$$

Note that the dimensionality of the state set can be different from the number of agents (i.e., $S \neq N$). State transitions are determined by a probability distribution over the future state, conditioned on the current state-action pair: $\boldsymbol{x}_{i+1} \sim p_{\boldsymbol{x}}(\cdot | x_i, a_i)$; where we use boldface notation for denoting random variables. State transitions can be equivalently expressed as a function, $f : \mathbb{X} \times \mathbb{A} \times \Theta \to \mathbb{X}$, that depends on some random variable $\boldsymbol{\theta}_i \in \Theta$, with distribution $p_{\boldsymbol{\theta}}(\cdot | \boldsymbol{x}_i, a_i)$, such that

$$\boldsymbol{x}_{i+1} = f(x_i, a_i, \boldsymbol{\theta}_i). \tag{2}$$

We include a vector of $C$ constraint functions, $g \triangleq (g^c)_{c=1}^C$, where $g^c : \mathbb{X} \times \mathbb{A} \mapsto \mathbb{R}$; and define the constraint sets for $i = 0$: $\mathbb{C}_0 \triangleq \mathbb{A} \cap \{a_0 : g(x_0, a_0) \leq 0\}$; and for $i = 0, \ldots, \infty$: $\mathbb{C}_i \triangleq \{\{\mathbb{X} \cap \{x_i : x_i = f(x_{i-1}, a_{i-1}, \theta_{i-1})\}\} \times \mathbb{A}\} \cap \{(x_i, a_i) : g(x_i, a_i) \leq 0\}$, which determine the feasible states and actions. The instantaneous reward of each agent, $\boldsymbol{r}_{k,i}$, is also a random variable conditioned on the current state-action pair: $\boldsymbol{r}_{k,i} \sim p_{\boldsymbol{r}_i}(\cdot | x_i, a_i)$. Given random variable $\boldsymbol{\sigma}_{k,i} \in \Sigma_k$ with distribution $p_{\boldsymbol{\sigma}_k}(\cdot | x_i, a_i)$, we define reward function $r_k : \mathbb{X} \times \mathbb{A} \times \Sigma_k \to \mathbb{R}$ for every agent $k \in \mathcal{N}$:

$$\boldsymbol{r}_{k,i} = r_k(x_i, a_i, \boldsymbol{\sigma}_{k,i}). \tag{3}$$

We assume that $\boldsymbol{\theta}_i$ and $\boldsymbol{\sigma}_{k,i}$ are independent of each other and of any other $\boldsymbol{\theta}_j$ and $\boldsymbol{\sigma}_{k,j}$, at every time step $j \neq i$, given $x_i$ and $a_i$.

Let $\pi_k : \mathbb{X} \to \mathbb{A}_k$ and $\pi : \mathbb{X} \to \mathbb{A}$ denote the policy for agent $k$ and all agents, respectively, such that:

$$a_{k,i} = \pi_k(x_i), \quad a_i = \pi(x_i), \quad \text{and} \quad \pi \triangleq (\pi_k)_{k \in \mathcal{N}}. \tag{4}$$

Let $\Omega_k$ and $\Omega = \prod_{k \in \mathcal{N}} \Omega_k$ denote the policy spaces for agent $k$ and for all agents, respectively, such that $\pi_k \in \Omega_k$ and $\pi \in \Omega$. Note that $\Omega(\mathbb{X}) = \mathbb{A}$. Introduce also $\pi_{-k} : \mathbb{X} \to \mathbb{A}_{-k}$ as the policy of all agents except that of agent $k$. Then, by slightly abusing notation, we write: $\pi = (\pi_k, \pi_{-k}), \forall k \in \mathcal{N}$.

The general (i.e., nonparametric) stochastic game with Markov dynamics consists in a multiobjective *variational* problem with design space $\Omega$ and objective space $\mathbb{R}^N$, where each agent aims to find a stationary policy that maximizes its expected discounted cumulative reward, for which the vector of constraints, $g$, is satisfied almost surely:

$$\mathcal{G}_1 : \forall k \in \mathcal{N} \quad \begin{array}{cl} \underset{\pi_k \in \Omega_k}{\text{maximize}} & \mathbb{E}\left[\sum_{i=0}^{\infty} \gamma^i r_k\left(\boldsymbol{x}_i, \pi_k(\boldsymbol{x}_i), \pi_{-k}(\boldsymbol{x}_i), \boldsymbol{\sigma}_i\right)\right] \\ \text{s.t.} & \boldsymbol{x}_{i+1} = f(\boldsymbol{x}_i, \pi(\boldsymbol{x}_i), \boldsymbol{\theta}_i), \\ & g(\boldsymbol{x}_i, \pi(\boldsymbol{x}_i)) \leq 0. \end{array} \tag{5}$$

Similar to static games, since there might not exist a policy that maximizes every agent's objective, we will rely on Nash equilibrium (NE) as solution concept. But rather than trying to find a variational NE solution for (5), we propose a more tractable approximate game by constraining the policies to belong to some finite-dimensional parametric family.

Introduce the set of parametric policies, $\Omega^w$, as a *finite*-dimensional function space with parameter $w \in \mathbb{W} \subseteq \mathbb{R}^W$: $\Omega^w \triangleq \{\pi(\cdot, w) : w \in \mathbb{W}\}$. Note that for a given $w$, the parametric policy is still

a mapping from states to actions: $\pi(\cdot, w) : \mathbb{X} \to \mathbb{A}$. Let $w_k \in \mathbb{W}_k \subseteq \mathbb{R}^{W_k}$ denote the parameter vector of length $W_k$ for the parametrized policy $\pi_k$, so that it lies in the finite-dimensional space $\Omega_k^w \triangleq \{\pi_k(\cdot, w_k) : w_k \in \mathbb{W}_k\}$, such that $\Omega^w \triangleq \prod_{k \in \mathcal{N}} \Omega_k^w$, $\mathbb{W} \triangleq \prod_{k \in \mathcal{N}} \mathbb{W}_k$, $W \triangleq \sum_{k \in \mathcal{N}} W_k$, and

$$w \triangleq (w_k)_{k \in \mathcal{N}}, \quad \pi(\cdot, w) \triangleq (\pi_k(\cdot, w_k))_{k \in \mathcal{N}}. \tag{6}$$

Let $w_{-k}$ denote the parameters of all agents except that of agent $k$, so that we can also write:

$$w = (w_k, w_{-k}), \quad \pi(\cdot, w) = \pi(\cdot, (w_k, w_{-k})) = (\pi_k(\cdot, w_k), \pi_{-k}(\cdot, w_{-k})). \tag{7}$$

In addition, we use $w_k(\ell)$ to denote the $\ell$-th component of $w_k$, such that $w_k \triangleq (w_k(\ell))_{\ell=1}^{W_k}$.

By constraining the policy of $\mathcal{G}_1$ to lie in $\Omega_k^w$, we obtain a multiobjective *optimization* problem with design space $\mathbb{W}$:

$$\begin{aligned} \mathcal{G}_2 \quad : \quad &\underset{w_k \in \mathbb{W}_k}{\text{maximize}} \quad \mathbb{E}\left[\sum_{i=0}^{\infty} \gamma^i r_k\left(\boldsymbol{x}_i, \pi_k(\boldsymbol{x}_i, w_k), \pi_{-k}(\boldsymbol{x}_i, w_{-k}), \boldsymbol{\sigma}_{k,i}\right)\right] \\ \forall k \in \mathcal{N} \quad &\text{s.t.} \quad \boldsymbol{x}_{i+1} = f(\boldsymbol{x}_i, \pi(\boldsymbol{x}_i, (w_k, w_{-k})), \boldsymbol{\theta}_i), \\ &\qquad g(\boldsymbol{x}_i, \pi(\boldsymbol{x}_i, (w_k, w_{-k}))) \leq 0. \end{aligned} \tag{8}$$

The solution concept in which we are interested is the *parametric closed-loop* Nash equilibrium (PCL-NE), which consists in a parametric policy for which no agent has incentive to deviate unilaterally.

**Definition 1** *A parametric closed-loop Nash equilibrium (PCL-NE) of $\mathcal{G}_2$ is a vector $w^\star = \left(w_k^\star, w_{-k}^\star\right) \in \mathbb{R}^W$ that satisfies:*

$$\mathbb{E}\left[\sum_{i=0}^{\infty} \gamma^i r_k\left(\boldsymbol{x}_i, \pi_k(\boldsymbol{x}_i, w_k^\star), \pi_{-k}(\boldsymbol{x}_i, w_{-k}^\star), \boldsymbol{\sigma}_{k,i}\right)\right]$$

$$\geq \mathbb{E}\left[\sum_{i=0}^{\infty} \gamma^i r_k\left(\boldsymbol{x}_i, \pi_k(\boldsymbol{x}_i, w_k), \pi_{-k}(\boldsymbol{x}_i, w_{-k}^\star), \boldsymbol{\sigma}_{k,i}\right)\right], \quad \forall k \in \mathcal{N}, \ \forall \boldsymbol{x}_0 = x_0 \in \mathbb{X},$$

$$\forall w_k \in \left\{w_k \in \mathbb{W}_k : a_i = \left(\pi_k(\boldsymbol{x}_i, w_k), \pi_{-k}(\boldsymbol{x}_i, w_{-k}^\star)\right) \in \mathbb{C}_0, (\boldsymbol{x}_i, a_i) \in \mathbb{C}_i, i = 1, \ldots, \infty\right\}. \tag{9}$$

Since $\mathcal{G}_2$ is similar to $\mathcal{G}_1$ but with an extra constraint on the policy set, loosely speaking, we can see a PCL-NE as a projection of some NE of $\mathcal{G}_1$ onto the manifold spanned by parametric family of choice. Hence, if the parametric family has arbitrary expressive capacity (e.g., a neural network with enough neurons in the hidden layers), we can expect that the resulting PCL-NE evaluated on $\mathcal{G}_1$ will approximate arbitrarily close the performance of an exact variational equilibrium.

We consider the following general assumptions.

**Assumption 1** *The state and parameter sets, $\mathbb{X}$ and $\mathbb{W}$, are nonempty and convex.*

**Assumption 2** *The reward functions $r_k$ are twice continuously differentiable in $\mathbb{X} \times \mathbb{W}$, $\forall k \in \mathcal{N}$.*

**Assumption 3** *The state-transition function, $f$, and constraints, $g$, are continuously differentiable in $\mathbb{X} \times \mathbb{W}$, and satisfy some regularity conditions (e.g., Mangasarian-Fromovitz).*

**Assumption 4** *The reward functions $r_k$ are proper, and there exists a scalar $B$ such that the level sets $\{a_0 \in \mathbb{C}_0, (x_i, a_i) \in \mathbb{C}_i : \mathbb{E}\left[r_k(x_i, a_i, \sigma_{k,i})\right] \geq B\}_{i=0}^{\infty}$ are nonempty and bounded $\forall k \in \mathcal{N}$.*

Assumptions 1–2 usually hold in engineering applications. Assumption 3 ensures the existence of feasible dual variables, which is required for establishing the optimality conditions. Assumption 4 will allow us to ensure the existence of PCL-NE. We say that $r_k$ is proper if: *i)* $\mathbb{E}\left[r_k(x_i, a_i, \boldsymbol{\sigma}_k)\right] > -\infty$ for at least one $(x_i, a_i) \in \mathbb{C}_i$, and *ii)* $\mathbb{E}\left[r_k(x_i, a_i, \boldsymbol{\sigma}_{k,i})\right] < \infty$, $\forall a_0 \in \mathbb{C}_0$, $\forall (x_i, a_i) \in \mathbb{C}_i$ $(i = 1, \ldots, \infty)$.

## 3 Standard Approach to Closed-Loop Markov Games

In this section, we review the standard approach for tackling CL dynamic games (González-Sánchez and Hernández-Lerma, 2013). For simplicity, we consider *deterministic* game and no constraints:

$$\mathcal{G}_{\text{std}} : \forall k \in \mathcal{N} \quad \underset{\pi_k \in \Omega_k}{\text{maximize}} \quad \sum_{i=0}^{\infty} \gamma^i r_k\left(x_i, \pi_k(x_i), \pi_{-k}(x_i)\right) \tag{10}$$
$$\text{s.t.} \quad x_{i+1} = f(x_i, \pi(x_i)).$$

First, it inverts $f$ to express the policy in reduced form, i.e., as a function of current and future states:

$$\pi(x_i) = h(x_i, x_{i+1}). \tag{11}$$

This implicitly assumes that such function $h : \mathbb{X} \times \mathbb{X} \to \mathbb{A}$ exists, which might not be the case if $f$ is not invertible. Next, $\pi_k$ is replaced with (11) in each $r_k$:

$$r_k\left(x_i, \pi_k(x_i), \pi_{-k}(x_i)\right) = r_k\left(x_i, h(x_i, x_{i+1})\right) \triangleq r'_k\left(x_i, x_{i+1}\right), \tag{12}$$

where $r'_k : \mathbb{X} \times \mathbb{X} \to \mathbb{R}$ is the reward in reduced-form. Then, the Euler equation (EE) and transversality condition (TC) are obtained from $r'_k$ for all $k \in \mathcal{N}$ and used as necessary optimality conditions:

$$\nabla_{x_i} r'_k\left(x_{i-1}, x_i\right) + \nabla_{x_i} r'_k\left(x_i, x_{i+1}\right) = 0 \qquad \text{(EE)}, \tag{13}$$
$$\lim_{i \to \infty} x_i^\top \nabla_{x_i} r'_k\left(x_{i-1}, x_i\right) = 0 \qquad \text{(TC)}. \tag{14}$$

When $r'_k$ are concave for all agents, and $\mathbb{X} \subseteq \mathbb{R}^+$ (i.e., $\mathbb{X} = \{x_i : x_i \geq 0, x_i \in \mathbb{R}^S\}$), these optimality conditions become sufficient for Nash equilibrium (González-Sánchez and Hernández-Lerma, 2013, Theorem 4.1). Thus, the standard approach consists in guessing parametric policies from the space of functions $\Omega$, and check whether any of these functions satisfies the optimality conditions. We illustrate this procedure with a well known resource-sharing game named "the great fish war" due to Levhari and Mirman (1980), with Example 1 in Appendix A.

Although the standard approach sketched above (see also Appendix A) has been the state-of-the-art for the analysis of CL dynamic games, it has some drawbacks: *i)* The reduced form might not exist; *ii)* constraints are not handled easily and we have to rely in *ad hoc* arguments for ensuring feasibility; *iii)* finding a specific parametric form that satisfies the optimality conditions can be extremely difficult since the space of functions is too large; and *iv)* the rewards have to be concave for all agents in order to guarantee that any policy that satisfies the conditions is an equilibrium.

In order to overcome these issues, we propose to first constrain the set of policies to some parametric family, and then derive the optimality conditions for this parametric problem; as opposed to the standard approach that first derives the optimality conditions of $\mathcal{G}_1$, and then guesses a parametric form that satisfies them. Based on this insight, we will introduce MPG with parametric policies as a class of games that can be solved with standard DRL techniques by finding the solution of a related (single-objective) OCP. We explain the details in the following section.

## 4 Closed-Loop Markov Potential Games

In this section, we extend the OL analysis of Zazo et al. (2016a) to the CL case. We define MPGs with CL information structure; introduce a parametric OCP; provide verifiable conditions for a parametric approximate game to be an MPG in the CL setting; show that when the game is an MPG, we can find a PCL-NE by solving the parametric OCP with a specific objective function; and provide a practical method for obtaining such objective function.

First, we define MPGs with CL information structure and parametric policies as follows.

**Definition 2** *Given a policy family $\pi(\cdot, w) \in \Omega^w$, game (8) is an MPG if and only if there is a function $J : \mathbb{X} \times \mathbb{W} \times \Sigma \to \mathbb{R}$, named the potential, that satisfies the following condition $\forall k \in \mathcal{N}$:*

$$\mathbb{E}\left[\sum_{i=0}^{\infty} \gamma^i \left(r_k(\boldsymbol{x}_i, \pi_k(\boldsymbol{x}_i, w_k), \pi_{-k}(\boldsymbol{x}_i, w_{-k}), \boldsymbol{\sigma}_{k,i}) - r_k(\boldsymbol{x}_i, \pi_k(\boldsymbol{x}_i, v_k), \pi_{-k}(\boldsymbol{x}_i, w_{-k}), \boldsymbol{\sigma}_{k,i})\right)\right]$$

$$= \mathbb{E}\left[\sum_{i=0}^{\infty} \gamma^i \left(J(\boldsymbol{x}_i, \pi_k(\boldsymbol{x}_i, w_k), \pi_{-k}(\boldsymbol{x}_i, w_{-k}), \boldsymbol{\sigma}_i) - J(\boldsymbol{x}_i, \pi_k(\boldsymbol{x}_i, v_k), \pi_{-k}(\boldsymbol{x}_i, w_{-k}), \boldsymbol{\sigma}_i)\right)\right],$$

$$\forall \boldsymbol{x}_i \in \mathbb{X}, \quad \forall w_k, \; v_k \in \mathbb{W}_k. \tag{15}$$

Definition 2 means that there exists some potential function, $J$, shared by all agents, such that if some agent $k$ changes its policy unilaterally, the change in its reward, $r_k$, equals the change in $J$.

The main contribution of this paper is to show that when (8) is a MPG, we can find one PCL-NE by solving a related parametric OCP. The generic form of such parametric OCP is as follows:

$$
\mathcal{P}_1 : \quad
\begin{aligned}
\underset{w \in \mathbb{W}}{\text{maximize}} \quad & \mathbb{E}\left[\sum_{i=0}^{\infty} \gamma^i J(\boldsymbol{x}_i, \pi(\boldsymbol{x}_i, w), \boldsymbol{\sigma}_i)\right] \\
\text{s.t.} \quad & \boldsymbol{x}_{i+1} = f(\boldsymbol{x}_i, \pi(\boldsymbol{x}_i, w), \boldsymbol{\theta}_i), \\
& g(\boldsymbol{x}_i, \pi(\boldsymbol{x}_i, w)) \leq 0.
\end{aligned}
\tag{16}
$$

where we replaced the multiple objectives (one per agent) with the potential $J$ as single objective. This is convenient since solving a single objective OCP is generally much easier than solving the Markov game. However, we still have to find out how to obtain $J$. The following Theorem formalizes the relationship between $\mathcal{G}_2$ and $\mathcal{P}_1$ and shows one way to obtain $J$ (proof in Appendix C).

**Theorem 1** *Let Assumptions 1–4 hold. Let the reward functions satisfy the following $\forall k, j \in \mathcal{N}$:*

$$
\mathbb{E}\left[\nabla_{w_j}\left[\nabla_{x_i} r_k(x_i, \pi(x_i, w), \boldsymbol{\sigma}_{k,i})\right]\right] = \mathbb{E}\left[\nabla_{w_k}\left[\nabla_{x_i} r_j(x_i, \pi(x_i, w), \boldsymbol{\sigma}_{j,i})\right]\right], \tag{17}
$$

$$
\mathbb{E}\left[\nabla_{x_i}\left[\nabla_{x_i} r_k(x_i, \pi(x_i, w), \boldsymbol{\sigma}_{k,i})\right]\right] = \mathbb{E}\left[\nabla_{x_i}\left[\nabla_{x_i} r_j(x_i, \pi(x_i, w), \boldsymbol{\sigma}_{j,i})\right]\right], \tag{18}
$$

$$
\mathbb{E}\left[\nabla_{w_j}\left[\nabla_{w_k} r_k(x_i, \pi(x_i, w), \boldsymbol{\sigma}_{k,i})\right]\right] = \mathbb{E}\left[\nabla_{w_k}\left[\nabla_{w_j} r_j(x_i, \pi(x_i, w), \boldsymbol{\sigma}_{j,i})\right]\right], \tag{19}
$$

*where the expected value is taken component-wise. Then, game (8) is an MPG that has a PCL-NE equal to the solution of OCP (16). The potential $J$ that is the instantaneous reward for the OCP is given by line integral:*

$$
\begin{aligned}
& J(x_i, \pi(x_i, w), \sigma_i) \\
&= \int_0^1 \sum_{k \in \mathcal{N}} \left( \sum_{m=1}^{S} \frac{\partial r_k\big(\eta(z), \pi_k(\eta(z), w_k), \pi_{-k}((\eta(z), w_{-k}), \sigma_{k,i}\big)}{\partial x_i(m)} \frac{d\eta_m(z)}{dz} \right. \\
& \qquad \left. + \sum_{\ell=1}^{A_k} \frac{\partial r_k\big(x_i, \pi_k(x_i, \xi_k(z)), \pi_{-k}(x_i, w_{-k}), \sigma_{k,i}\big)}{\partial a_{k,i}(\ell)} \frac{d\xi_{k,\ell}(z)}{dz} \right) dz,
\end{aligned}
\tag{20}
$$

*where $\eta(z) \triangleq (\eta_k(z))_{m=1}^{S}$ and $\xi(z) \triangleq (\xi_k(z))_{k \in \mathcal{N}}$ are piecewise smooth paths in $\mathbb{X}$ and $\mathbb{W}$, respectively, with components $\xi_k(z) \triangleq (\xi_{k,\ell}(z))_{\ell=1}^{W_k}$, such that the initial and final state-action conditions are given by $(\eta(0), \xi(0))$ and $(\eta(1) = x_i, \xi(1) = w)$.*

From (20), we can see that $J$ is obtained through the line integral of a vector field with components the partial derivatives of the agents' rewards (see Appendix C), and so the name *potential* function. Note also that Theorem 1 proves that any solution to $\mathcal{P}_1$ is also a PCL-NE of $\mathcal{G}_2$, but we remark that there may be more equilibria of the game that are not solutions to $\mathcal{P}_1$ (see Appendix C).

The usefulness of Theorem 1 is that, once we have the potential function, we can formulate and solve the related OCP for any specific parametric policy family. This is a considerable improvement over the standard approach. On one hand, if the chosen parametric policy contains the optimal solution, then we will obtain the same equilibrium as the standard approach. On the other hand, if the chosen parametric family does not have the optimal solution, the standard approach will fail, while our approach will always provide a solution that is an approximation (a projection over $\Omega^w$) of an exact variational equilibrium. Moreover, as mentioned above, we can expect that the more expressive the parametric family, the more accurate the approximation to the variational equilibrium. In Appendix B, we show how to to solve "the great fish war" game with the proposed framework, yielding the same solution as with the standard approach, with no loss of accuracy.

Although expressing $J$ as a line integral of a field is theoretically appealing, if the parametric family is involved—as it is usually the case for expressive policies like deep neural-networks—then (20) might be difficult to evaluate. The following results show how to obtain $J$ easily by visual inspection.

First, the following corollary follows trivially from (17)–(19) and shows that *cooperative* games, where all agents have the same reward, are MPGs, and the potential equals the reward:

**Corollary 1** *Cooperative games, where all agents have a common reward, such that*

$$r_k(x_i, \pi(x_i, w), \sigma_{k,i}) = J(x_i, \pi(x_i, w), \sigma_i), \quad \forall k \in \mathcal{N}, \qquad (21)$$

*are MPGs; and the potential function* (20) *equals the common reward function in* (21).

Second, we address noncooperative games, and show that the potential can be found by inspection as a separable term that is common to all agents' reward functions. Interestingly, we will also show that a game is an MPG in the CL setting if and only if all agents' policies depend on *disjoint subsets* of components of the state vector. More formally, introduce $\mathcal{X}_k^\pi$ as the set of state vector components that influence the policy of agent $k$ and introduce a new state vector, $x_k^\pi$, and let $x_{-k,i}^\pi$ be the vector of components that do not influence the policy of agent $k$:

$$x_{k,i}^\pi \triangleq (x_i(m))_{m \in \mathcal{X}_k^\pi}, \quad x_{-k,i}^\pi \triangleq (x_i(l))_{l \notin \mathcal{X}_k^\pi}. \qquad (22)$$

In addition, introduce $\mathcal{X}_k^r$ as the set of components of the state vector that influence the reward of agent $k$ directly (not indirectly through any other agent's policy), and define the state vectors:

$$x_{k,i}^r \triangleq (x_i(m))_{m \in \mathcal{X}_k^r}, \quad x_{-k,i}^r \triangleq (x_i(l))_{l \notin \mathcal{X}_k^r}. \qquad (23)$$

Introduce also the union of these two subsets, $\mathcal{X}_k^\Theta = \mathcal{X}_k^\pi \cup \mathcal{X}_k^r$, and its corresponding vectors:

$$x_{k,i}^\Theta \triangleq (x_i(m))_{m \in \mathcal{X}_k^\Theta}, \quad x_{-k,i}^\Theta \triangleq (x_i(m))_{m \notin \mathcal{X}_k^\Theta}. \qquad (24)$$

Then, the following theorem allows us to obtain the potential function (proof in Appendix D).

**Theorem 2** *Let Assumptions 1–4 hold. Then, game* (8) *is an MPG if and only if: i) the reward function of every agent can be expressed as the sum of a term common to all agents plus another term that depends neither on its own state-component vector, nor on its policy parameter:*

$$r_k\left(x_{k,i}^r, \pi(x_{k,i}^\pi, w_k), \pi(x_{-k,i}^\pi, w_{-k}), \sigma_{k,i}\right) = J\left(x_i, \pi(x_i, w), \sigma_i\right)$$
$$+ \Theta_k\left(x_{-k,i}^r, \pi(x_{-k,i}^\pi, w_{-k}), \sigma_i\right), \quad \forall k \in \mathcal{N}; \quad (25)$$

*and ii) the following condition on the non-common term holds:*

$$\mathbb{E}\left[\nabla_{x_{k,i}^\Theta} \Theta_k\left(x_{-k,i}^r, \pi(x_{-k,i}^\pi, w_{-k}), \boldsymbol{\sigma}_i\right)\right] = 0. \qquad (26)$$

*Moreover, if* (26) *holds, then the common term in* (25), *J, equals the potential function* (20).

Note that (26) holds in the following cases: *i)* when $\Theta_k = 0$, as the cooperative case described in Corollary 1; *ii)* when $\Theta_k$ does not depend on the state but only on the parameter vector, i.e., $\Theta_k : \prod_{j \in \mathcal{N}, j \neq k} \mathbb{W}_j \mapsto \mathbb{R}$, as in "the great fish war" example described in Appendix B; or *iii)* when all agents have disjoint state-component subsets, i.e., $\mathcal{X}_k^\Theta \cap \mathcal{X}_j^\Theta = \emptyset, \forall(k, j) \in \{\mathcal{N} \times \mathcal{N} : k \neq j\}$.

An interesting insight from Theorem 2 is that a dynamic game that is potential when it is analyzed in the OL case (i.e., the policy is a predefined sequence of actions), might not be potential when analyzed in the CL parametric setting. This conclusion is straightforward since the potentiality condition in the OL case provided by (Valcarcel Macua et al., 2016, Cor. 1) is equal to (25), without requiring (26).

In order to apply Theorems 1 and 2, we are implicitly assuming that there exists solution to the OCP. We finish this section, by showing that this is actually the case in our setting (proof in Appendix E).

**Proposition 1** *Under Assumption 4, OCP* (16) *has nonempty solution set.*

In other words, Prop. 1 shows that there exists a deterministic policy that achieves the optimal value of $\mathcal{P}_1$, which is also an NE of $\mathcal{G}_2$ if conditions (17)–(19) or equivalently (25)–(26) hold. We remark that there might be many other—possibly stochastic—policies that are also NE of the game.

## 5 EXPERIMENT

In this section, we show how to use the proposed MPGs framework to learn an equilibrium of a communications engineering application. We extend the Medium Access Control (MAC) game presented in (Zazo et al., 2016a) to stochastic dynamics and rewards (where previous OL solutions

would fail), and use the Trust Region Policy Optimization (TRPO) algorithm (Schulman et al., 2015), which is a reliable reinforcement learning method policy search method that approximates the policy with a deep-neural network, to learn a policy that is a PCL-NE of the game.

We consider a MAC uplink scenario with $N = 4$ agents, where each agent is a user that sets its transmitter power aiming to maximize its data rate and battery lifespan. If multiple users transmit at the same time, they will interfere with each other and decrease their rate, using their batteries inefficiently, so that they have to find an equilibrium. Let $x_{k,i} \in [0, B_{k,\max}] \triangleq \mathbb{X}_k$ denote the battery level for each agent $k \in \mathcal{N}$, which is discharged proportionally to the transmitted power, Let $a_{k,i} \in [0, P_{k,\max}] \triangleq \mathbb{A}_k$ be the transmitted power for the $k$-th user, where constants $P_{k,\max}$ and $B_{k,\max}$ stand for the maximum allowed transmitter power and battery level, respectively. The system state is the vector with all user's battery levels: $x_i = (x_{k,i})_{k \in \mathcal{N}} \in \mathbb{X}$; such that $S = N$ and all state vector components are unshared, i.e., $\mathbb{X} = \prod_{k \in \mathcal{N}} \mathbb{X}_k \subset \mathbb{R}^N$, and $\mathcal{X}_k = \{k\}$. We remark that although each agent's battery depletion level depends directly on its action and its previous battery level only, it also depends indirectly on the strategies and battery levels of the rest of agents. The game can be formalized as follows:

$$\mathcal{G}_{\text{mac}} : \quad \underset{w_k \in \mathbb{A}}{\text{maximize}} \quad \sum_{i=0}^{\infty} \gamma^i \left( \log \left( 1 + \frac{|\boldsymbol{h}_{k,i}|^2 \pi(\boldsymbol{x}_{k,i}, w_k)}{1 + \sum_{j \in \mathcal{N}:j \neq k} |\boldsymbol{h}_j|^2 \pi(\boldsymbol{x}_{j,i}, w_j)} \right) + \alpha \boldsymbol{x}_{k,i} \right)$$

$$\forall k \in \mathcal{N} \qquad \text{s.t.} \quad \boldsymbol{x}_{k,i+1} = \boldsymbol{x}_{k,i} - \boldsymbol{\delta}_i \pi(\boldsymbol{x}_{k,i}, w_k), \quad x_{k,0} = B_{k,\max} \qquad (27)$$

$$0 \leq \pi(\boldsymbol{x}_{k,i}, w_k) \leq P_{k,\max}, \quad 0 \leq \boldsymbol{x}_{k,i} \leq B_{k,\max}, \quad i = 0, \dots, \infty$$

where $\boldsymbol{h}_k$ is the random fading channel coefficient for user $k$, $\alpha$ is the weight for the battery reward term, and $\delta$ is the discharging factor.

First of all, note that each agent's policy and reward depend only on its own battery level, $\boldsymbol{x}_{k,i}$. Therefore, we can apply Theorem 2 and establish that the game is a MPG, with potential function:

$$J(x_i, \pi(x_i, w)) = \log \left( 1 + \sum_{k \in \mathcal{N}} |h_{k,i}|^2 \pi(x_{k,i}, w_k) \right) + \alpha \sum_{k \in \mathcal{N}} x_{k,i} \qquad (28)$$

Thus, we can formulate OCP (16) with single objective given by (28).

Since the battery level is a positive term in the reward, the optimal policy will make the battery deplete in finite time (formal argument can be derived from transversality condition (54)). Moreover, since $\boldsymbol{\delta}_{k,i} \geq 0$, the episode gets into a stationary (i.e., terminal) state once the battery has been depleted. We have chosen the reward to be convex. The reason is that in order to compute a benchmark solution, we can solve the finite time-horizon convex OCP exactly with a convex optimization solver, e.g., CVX (Grant and Boyd, 2014), and use the result as a baseline for comparing with the solution learned by a DRL algorithm. Nevertheless, standard solvers do not allow to include random variables. To surmount this issue, we generated 100 independent sequences of samples of $\boldsymbol{h}_{k,i}$ and $\boldsymbol{\delta}_{k,i}$ for all $k \in \mathcal{N}$ and length $T = 100$ time steps each, and obtain two solutions with them. We set $|\boldsymbol{h}_{k,i}|^2 = |h_k|^2 \boldsymbol{v}_{k,i}$, where $\boldsymbol{v}_{k,i}$ is uniform in $[0.5, 1]$, $|h_1|^2 = 2.019$, $|h_2|^2 = 1.002$, $|h_3|^2 = 0.514$ and $|h_4|^2 = 0.308$; and $\boldsymbol{\delta}_{k,i}$ is uniform in $[0.7, 1.3]$. The first solution is obtained by averaging the sequences, and building a deterministic convex problem with the average sequence, which yielded an optimal value $V^\star_{\text{cvx}} = 33.19$. We consider $V^\star_{\text{cvx}}$ to be an estimator of the optimal value of the stochastic OCP. The second solution is obtained by building 100 deterministic problems, solving them, and averaging their optimal values, which yielded an optimal value $V^\star_{\text{avg,cvx}} = 34.90$. We consider $V^\star_{\text{avg,cvx}}$ to be an upper bound estimate of the optimal value of the stochastic OCP (Jensen's inequality). The batteries depleted at a level $x_T < 10^{-6}$ in all cases, concluding that time horizon of $T = 100$ steps is valid. We remark that these benchmark solutions required complete knowledge of the game.

When we have no prior knowledge of the dynamics and rewards, the proposed approach allows as to learn a PCL-NE of (27) by using any DRL method that is suitable for continuous state and actions, like TRPO (Schulman et al., 2015), DDPG (Lillicrap et al., 2015) or A3C (Mnih et al., 2016). DRL methods learn by interacting with a black-box simulator, such that at every time step $i$, agents observe state $x_i$, take action $a_i = \pi_w(x_i)$ and observe the new stochastic battery levels and reward values, with no prior knowledge of the reward or state-dynamic functions.

As a proof of concept, we perform simulations with TRPO, approximating the policy with a neural network with 3 hidden layers of size 32 neurons per layer and RELU activation function, and an

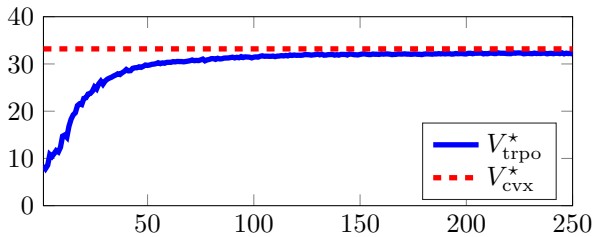

Figure 1: Results for the MAC game (27) obtained with TRPO and the averaged solutions given by the convex optimization solver).

output layer that is the mean of a Gaussian distribution. Each iteration of TRPO uses a batch of size $4000$ simulation steps (i.e., tuples of state transition, action and rewards). The step-size is $0.01$. Figure 1 shows the results. After $400$ iterations, TRPO achieves an optimal value $V_{\text{trpo}}^{\star} = 32.34$, which is $97.44\%$ of $V_{\text{cvx}}^{\star}$, and $92.7\%$ of the upper bound $V_{\text{avg,cvx}}^{\star}$.

## 6 CONCLUSIONS

We have extended previous results on MPGs with constrained continuous state-action spaces providing practical conditions and a detailed analysis of Nash equilibrium with parametric policies, showing that a PCL-NE can be found by solving a related OCP. Having established a relationship between a MPG and an OCP is a significant step for finding an NE, since we can apply standard optimal control and reinforcement learning techniques. We illustrated the theoretical results by applying TRPO (a well known DRL method) to an example engineering application, obtaining a PCL-NE that yields near optimal results, very close to an exact variational equilibrium.

## 7 ACKNOWLEDGEMENTS

We thank David Mguni, Enrique Munoz de Cote, and Haitham Bou-Ammar for insightful discussions.

This work was partially supported by the Spanish Ministry of Science and Innovation under the grant TEC2016-76038-C3-1-R (HERAKLES) and the COMONSENS Network of Excellence TEC2015-69648-REDC.

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

## A   EXAMPLE: THE "GREAT FISH WAR" GAME – STANDARD APPROACH

Let us illustrate the standard approach described in Section 3 with a well known resource-sharing game named "the great fish war" due to Levhari and Mirman (1980). We follow (González-Sánchez and Hernández-Lerma, 2013, Sec. 4.2).

**Example 1.**   Let $x_i$ be the stock of fish at time $i$, in some fishing area. Suppose there are $N$ countries obtaining reward from fish consumption, so that they aim to solve the following game:

$$
\mathcal{G}_{\text{fish}}: \quad
\begin{array}{c}
\underset{\pi_k \in \Omega_k}{\text{maximize}} \quad \sum_{i=0}^{\infty} \gamma^i \log\left(\pi_k(x_i)\right) \\
\forall k \in \mathcal{N} \\
\text{s.t.} \quad x_{i+1} = \left(x_i - \sum_{k \in \mathcal{N}} \pi_k(x_i)\right)^{\alpha}, \quad x_i \geq 0, \ \pi_k(x_i) \geq 0, \ i = 0, \ldots, \infty,
\end{array}
\tag{29}
$$

where $x_0 \geq 0$ and $0 < \alpha < 1$ are given.

In order to solve $\mathcal{G}_{\text{fish}}$, let us express each agent's action as:

$$
\pi_k(x_i) = x_i - x_{i+1}^{1/\alpha} - \sum_{j \in \mathcal{N}: j \neq k} \pi_j(x_i),
\tag{30}
$$

so that the rewards can be also expressed in reduced form, as required by the standard-approach:

$$
r_k'(x_i) = \log\left(x_i - x_{i+1}^{1/\alpha} - \sum_{j \in \mathcal{N}: j \neq k} \pi_j(x_i)\right).
\tag{31}
$$

Thus, the Euler equations for every agent $k \in \mathcal{N}$ and all $t = 0, \ldots, \infty$ become:

$$
\frac{-x_i^{1/\alpha - 1}/\alpha}{x_{i-1} - x_i^{1/\alpha} - \sum_{j \in \mathcal{N}: j \neq k} \pi_j(x_{i-1})} + \gamma \frac{1 - \sum_{j \in \mathcal{N}: j \neq k} \partial \pi_j(x_i)/\partial x_i}{x_i - x_{i+1}^{1/\alpha} - \sum_{j \in \mathcal{N}: j \neq k} \pi_j(x_i)} = 0.
\tag{32}
$$

Now, the standard method consists in guessing a family of parametric functions that replaces the policy, and checking whether such parametric policy satisfies (32) for some parameter vector. Let us try with policies that are linear mappings of the state:

$$
\pi_k(x_i) = w_k x_i.
\tag{33}
$$

By replacing (33) in (32), we obtain the following set of equations:

$$
\alpha \gamma \left(1 + w_k - \sum_{j \in \mathcal{N}} w_j\right) = 1 - \sum_{j \in \mathcal{N}} w_j, \quad \forall k \in \mathcal{N}.
\tag{34}
$$

Fortunately, it turns out that (34) has solution (which might not be the case for other policy parametrization), with parameters given by:

$$
w_k = \frac{1 - \alpha\gamma}{\alpha\gamma + N(1 - \alpha\gamma)}, \quad \forall k \in \mathcal{N}.
\tag{35}
$$

Since $0 < \alpha < 1$ and $0 \leq \gamma < 1$, it is apparent that $w_k > 0$ and the constraint $\pi_k(x_i) \geq 0$ holds for all $x_i \geq 0$. Moreover, since $\sum_{k \in \mathcal{N}} w_k < 1$, we have that $x_{i+1} \geq 0$ for any $x_0 \geq 0$. In addition, since $x_i$ is a resource and the actions must be nonnegative, it follows that $\lim_{i \to \infty} x_i = 0$ (there is no reason to save some resource). Therefore, the transversality condition holds. Since the rewards are concave, the states are non-negative and the linear policies with these coefficients satisfy the Euler and transversality equations, we conclude that they constitute an equilibrium (González-Sánchez and Hernández-Lerma, 2013, Theorem 4.1). $\triangle$

## B   EXAMPLE: "GREAT FISH WAR" GAME – PROPOSED APPROACH

In this section, we illustrate how to apply the proposed approach with the same "the great fish war" example, obtaining the same results as with the standard approach.

**Example 2.**   Consider "the great fish war" game described in Example 1. In order to use our approach, we replace the generic policy with the specific policy mapping of our preference. We

choose the linear mapping, $\pi_k(x_i) = w_k x_i$, to be able to compare the results with those obtained with the standard approach. Thus, we have the following game:

$$
\mathcal{G}_{\text{fish},w} : \\
\forall k \in \mathcal{N}
\qquad
\begin{aligned}
&\underset{w_k \in \mathbb{W}_k}{\text{maximize}} \quad \sum_{i=0}^{\infty} \gamma^i \log\left(w_k x_i\right) \\
&\text{s.t.} \quad x_{i+1} = \left( x_i - \sum_{k \in \mathcal{N}} w_k x_i \right)^\alpha, \\
&\qquad\quad x_i \geq 0, \quad w_k x_i \geq 0, \quad i = 0, \ldots, \infty.
\end{aligned}
\tag{36}
$$

Let us verify conditions (67)–(68). For all $k, j \in \mathcal{N}$ we have:

$$
r_k\left(x_i, \pi(x_i, w)\right) = \log\left(w_k x_i\right), \tag{37}
$$
$$
\nabla_{x_i} r_k\left(x_i, \pi(x_i, w)\right) = 1/x_i, \tag{38}
$$
$$
\nabla_{w_k} r_k\left(x_i, \pi(x_i, w)\right) = 1/w_k, \tag{39}
$$
$$
\nabla_{x_i}\nabla_{x_i} r_k\left(x_i, \pi(x_i, w)\right) = \nabla_{x_i}\nabla_{x_i} r_j\left(x_i, \pi(x_i, w)\right) = -1/x_i^2, \tag{40}
$$
$$
\nabla_{w_j}\nabla_{x_i} r_k\left(x_i, \pi(x_i, w)\right) = \nabla_{w_k}\nabla_{x_i} r_j\left(x_i, \pi(x_i, w)\right) = 0, \tag{41}
$$
$$
\nabla_{w_j}\nabla_{w_k} r_k\left(x_i, \pi(x_i, w)\right) = \nabla_{w_k}\nabla_{w_j} r_j\left(x_i, \pi(x_i, w)\right) = 0. \tag{42}
$$

Since conditions (67)–(68) hold, we conclude that (36) is an MPG. By applying the line integral (20), we obtain:

$$
J(x_i, w_i) = \log(x_i) + \sum_{k \in \mathcal{N}} \log\left(w_k\right). \tag{43}
$$

Now, we can solve OCP (16) with potential function (43). For this particular problem, it is easy to solve the KKT system in closed form. Introduce a shorthand:

$$
\overline{w} \triangleq \sum_{k \in \mathcal{N}} w_k. \tag{44}
$$

The Euler-Lagrange equation (62) for this problem becomes:

$$
\gamma^i + \beta_i \alpha x_i^\alpha \left(1 - \overline{w}\right)^\alpha - \beta_{i-1} x_i = 0. \tag{45}
$$

The optimality condition (64) with respect to the policy parameter becomes:

$$
\gamma^i - \beta_i \alpha x_i^\alpha \left(1 - \overline{w}\right)^{\alpha-1} w_k = 0. \tag{46}
$$

Let us solve for $\beta_i$ in (46):

$$
\beta_i = \frac{\gamma^i}{\alpha x_i^\alpha \left(1 - \overline{w}\right)^{\alpha-1} w_k}. \tag{47}
$$

Replacing (47) and the state-transition dynamics in (45), we obtain the following set of equations:

$$
\alpha\gamma\left(1 + w_k - \overline{w}\right) = 1 - \overline{w}, \quad \forall k \in \mathcal{N}. \tag{48}
$$

Hence, the parameters can be obtained as:

$$
w_k = \frac{1 - \alpha\gamma}{\alpha\gamma + N(1 - \alpha\gamma)}, \quad \forall k \in \mathcal{N}. \tag{49}
$$

This is exactly the same solution that we obtained in Example 1 with the standard approach. We remark that for the standard approach, we were able to obtain the policy parameters since we put the correct parametric form of the policy in the Euler equation. If we had used another parametric family without a linear term, the Euler equations (32) might have no solution and we would have got stuck. In contrast, with our approach, we could freely choose any other form of the parametric policy, and always solve the KKT system of the approximate game. Broadly speaking, we can say that the more expressive the parametric family, the more likely that the optimal policy of the original game will be accurately approximated by the optimal solution of the approximate game. $\triangle$

## C   PROOF OF THEOREM 1

*Proof:* The proof mimics the OL analysis from Zazo et al. (2016a). Let us build the KKT systems for the game and the OCP with parametric policies. For game (8), each agent's Lagrangian is given $\forall k \in \mathcal{N}$ by

$$\mathcal{L}_k\big(\boldsymbol{x}_{0:\infty}, w, \boldsymbol{\sigma}_{k,0:\infty}, \boldsymbol{\theta}_{0:\infty}, \boldsymbol{\lambda}_{k,0:\infty}, \boldsymbol{\mu}_{k,0:\infty}\big) = \mathbb{E}\Bigg[\sum_{i=0}^{\infty} \gamma^i \Big( r_k\,(\boldsymbol{x}_i, \pi(\boldsymbol{x}_i, w), \boldsymbol{\sigma}_{k,i})$$
$$+ \boldsymbol{\lambda}_{k,i}^\top \,(f\,(\boldsymbol{x}_i, \pi(\boldsymbol{x}_i, w), \boldsymbol{\theta}_i) - \boldsymbol{x}_{i+1}) + \boldsymbol{\mu}_{k,i}^\top \, g\,(\boldsymbol{x}_i, \pi(\boldsymbol{x}_i, w)) \Big) \Bigg], \quad (50)$$

where $\boldsymbol{\lambda}_{k,i} \triangleq (\boldsymbol{\lambda}_{k,s,i})_{s=1}^S \in \mathbb{R}^S$ and $\boldsymbol{\mu}_{k,i} \triangleq (\boldsymbol{\mu}_{k,c,i})_{c=1}^C \in \mathbb{R}^C$ are the vectors of multipliers at time $i$ (which are random since they depend on $\boldsymbol{\theta}_i$ and $\boldsymbol{x}_i$), and we introduced:

$$\boldsymbol{x}_{0:\infty} \triangleq (\boldsymbol{x}_i)_{i=0}^{\infty}, \quad a_{0:\infty} \triangleq (a_i)_{i=0}^{\infty}, \quad \boldsymbol{\lambda}_{k,0:\infty} \triangleq (\boldsymbol{\lambda}_{k,i})_{i=0}^{\infty}, \quad \boldsymbol{\mu}_{k,0:\infty} \triangleq (\boldsymbol{\mu}_{k,i})_{i=0}^{\infty}. \quad (51)$$

Introduce a shorthand for the instantaneous Lagrangian of agent $k$:

$$\Phi_k\big(x_i, \boldsymbol{x}_{i+1}, w, \boldsymbol{\sigma}_{k,i}, \boldsymbol{\theta}_i, \boldsymbol{\lambda}_{k,i}, \mu_{k,i}\big) \triangleq \mathbb{E}\Big[ r_k\,(x_i, \pi(x_i, w), \boldsymbol{\sigma}_{k,i})$$
$$+ \boldsymbol{\lambda}_{k,i}^\top\,(f\,(x_i, \pi(x_i, w), \boldsymbol{\theta}_i) - \boldsymbol{x}_{i+1}) + \mu_{k,i}^\top\, g\,(x_i, \pi(x_i, w)) \Big]. \quad (52)$$

The discrete time stochastic Euler-Lagrange equations applied to each agent's Lagrangian are different from the OL case studied in Zazo et al. (2016a) (see also (Sage and White, 1977, Sec. 6.1)), since we only take into account the *variation* with respect to the state:

$$\mathbb{E}\big[\nabla_{x_i}\Phi_k(x_i, \boldsymbol{x}_{i+1}, w, \boldsymbol{\sigma}_{k,i}, \boldsymbol{\theta}_i, \boldsymbol{\lambda}_{k,i}, \mu_{k,i})\big]$$
$$+ \nabla_{x_i}\big[\Phi_k\,(x_{i-1}, \boldsymbol{x}_i, w, \sigma_{k,i-1}, \theta_{i-1}, \lambda_{k,i-1}, \mu_{k,i-1})\big] = 0_S, \quad i = 1, \dots, \infty, \quad (53)$$

where $0_S$ denotes the vector of length $S$. The transversality condition is given by

$$\lim_{i \to \infty} \mathbb{E}\big[x_{k,i}^\top \nabla_{x_{k,i}}\Phi_k\,(x_i, \boldsymbol{x}_{i+1}, w, \boldsymbol{\sigma}_{k,i}, \boldsymbol{\theta}_i, \boldsymbol{\lambda}_{k,i}, \mu_{k,i})\big] = 0_S. \quad (54)$$

In addition, we have an optimality condition for the policy parameter $w_k$:

$$\mathbb{E}\big[\nabla_{w_k}\Phi_k\,(x_i, \boldsymbol{x}_{i+1}, w, \boldsymbol{\sigma}_{k,i}, \boldsymbol{\theta}_i, \boldsymbol{\lambda}_{k,i}, \mu_{k,i})\big] = 0_{W_k}. \quad (55)$$

From these first-order optimality conditions, we obtain the KKT system for every agent $k \in \mathcal{N}$ and all time steps $i = 1, \dots, \infty$:

$$\mathbb{E}\Big[\nabla_{x_i}\big[r_k\,(x_i, \pi(x_i, w), \boldsymbol{\sigma}_{k,i}) + \boldsymbol{\lambda}_{k,i}^\top\, f\,(x_i, \pi(x_i, w), \boldsymbol{\theta}_i)\big]\Big]$$
$$+ \nabla_{x_i}\big[\mu_{k,i}^\top\, g\,(x_i, \pi(x_i, w))\big] - \lambda_{k,i-1} = 0_{S_k}, \quad (56)$$

$$\lim_{i \to \infty} \mathbb{E}\Big[x_{k,i}^\top \nabla_{x_i}\big[r_k\,(x_i, \pi(x_i, w), \boldsymbol{\sigma}_{k,i}) + \boldsymbol{\lambda}_{k,i}^\top\, f\,(x_i, \pi(x_i, w), \boldsymbol{\theta}_i)\big]\Big]$$
$$+ \nabla_{x_i}\big[\mu_{k,i}^\top\, g\,(x_i, \pi(x_i, w))\big] = 0_{S_k}, \quad (57)$$

$$\mathbb{E}\Big[\nabla_{w_k}\big[r_k\,(x_i, \pi(x_i, w), \boldsymbol{\sigma}_{k,i}) + \boldsymbol{\lambda}_{k,i}^\top\, f\,(x_i, \pi(x_i, w), \boldsymbol{\theta}_i)\big]\Big]$$
$$+ \nabla_{w_k}\big[\mu_{k,i}^\top\, g\,(x_i, \pi(x_i, w))\big] = 0_{W_k}, \quad (58)$$

$$\boldsymbol{x}_{i+1} = f\,(x_i, \pi(x_i, w), \boldsymbol{\theta}_i), \quad g\,(x_i, \pi(x_i, w)) \leq 0_C, \quad (59)$$

$$\mu_{k,i} \leq 0_C, \quad \mu_{k,i}^\top\, g\,(x_i, \pi(x_i, w)) = 0, \quad (60)$$

where $\lambda_{k,i-1}$ is considered deterministic since it is known at time $i$.

Now, we derive the KKT system of optimality conditions for the OCP (16). The Lagrangian for (16) is given by:

$$\mathcal{L}^{\mathrm{OCP}}\big(\boldsymbol{x}_{0:\infty}, w, \boldsymbol{\sigma}_{k,0:\infty}, \boldsymbol{\theta}_{0:\infty}, \boldsymbol{\beta}_{0:\infty}, \boldsymbol{\delta}_{0:\infty}\big) = \mathbb{E}\Bigg[\sum_{i=0}^{\infty} \gamma^i \Big( J\,(\boldsymbol{x}_i, \pi(x_i, w), \boldsymbol{\sigma}_i)$$
$$+ \boldsymbol{\beta}_i^\top\,(f\,(\boldsymbol{x}_i, \pi(x_i, w), \boldsymbol{\theta}_i) - \boldsymbol{x}_{i+1}) + \boldsymbol{\delta}_i^\top\, g\,(\boldsymbol{x}_i, \pi(x_i, w)) \Big) \Bigg], \quad (61)$$

where $\boldsymbol{\beta}_i \triangleq (\boldsymbol{\beta}_{k,s,i})_{s=1}^S \in \mathbb{R}^S$ and $\boldsymbol{\delta}_i \triangleq (\boldsymbol{\delta}_{k,c,i})_{c=1}^C \in \mathbb{R}^C$ are the corresponding multipliers, which are random variables since they depend on $\boldsymbol{\theta}_i$ and $\boldsymbol{x}_i$. By taking the discrete time stochastic Euler-Lagrange equations and the optimality condition with respect to the policy parameter for the OCP, we obtain are a KKT system for the OCP: $i = 1, \ldots, \infty$:

$$\mathbb{E}\Big[\nabla_{x_i}\big[J(x_i, \pi(x_i, w), \boldsymbol{\sigma}_i) + \boldsymbol{\beta}_i^\top f(x_i, \pi(x_i, w), \boldsymbol{\theta}_i)\big]\Big]$$
$$+ \nabla_{x_i}\big[\delta_i^\top g(x_i, \pi(x_i, w))\big] - \beta_{i-1} = 0_{S_k}, \tag{62}$$

$$\lim_{i \to \infty} \mathbb{E}\Big[x_i^\top \nabla_{x_i}\big[J(x_i, \pi(x_i, w), \boldsymbol{\sigma}_i) + \boldsymbol{\beta}_i^\top f(x_i, \pi(x_i, w), \boldsymbol{\theta}_i)\big]\Big]$$
$$+ \nabla_{x_i}\big[\delta_i^\top g(x_i, \pi(x_i, w))\big] = 0_{S_k}, \tag{63}$$

$$\mathbb{E}\Big[\nabla_w\big[J(x_i, \pi(x_i, w), \boldsymbol{\sigma}_i) + \boldsymbol{\beta}_i^\top f(x_i, \pi(x_i, w), \boldsymbol{\theta}_i)\big]\Big]$$
$$+ \nabla_{w_k}\big[\delta_i^\top g(x_i, \pi(x_i, w))\big] = 0_A, \tag{64}$$

$$\boldsymbol{x}_{i+1} = f(x_i, \pi(x_i, w), \boldsymbol{\theta}_i), \quad g(x_i, \pi(x_i, w)) \le 0_C, \tag{65}$$

$$\delta_i \le 0_C, \quad \delta_i^\top g(x_i, \pi(x_i, w)) = 0, \tag{66}$$

where $\beta_{i-1}$ is known at time $i$ and includes the multipliers related to $x_{i-1}$.

By comparing (56)–(60) and (62)–(66), we conclude that both KKT systems are equal if the following holds $\forall k \in \mathcal{N}$ and $i = 1, \ldots, \infty$:

$$\mathbb{E}[\nabla_{x_i} r_k(x_i, \pi(x_i, w), \boldsymbol{\sigma}_{k,i})] = \mathbb{E}[\nabla_{x_i} J(x_i, \pi(x_i, w), \boldsymbol{\sigma}_i)], \tag{67}$$

$$\mathbb{E}[\nabla_{w_k} r_k(x_i, \pi(x_i, w), \boldsymbol{\sigma}_{k,i})] = \mathbb{E}[\nabla_{w_k} J(x_i, \pi(x_i, w), \boldsymbol{\sigma}_i)], \tag{68}$$

$$\boldsymbol{\lambda}_{k,i} = \boldsymbol{\beta}_i, \quad \mu_{k,i} = \delta_i. \tag{69}$$

Since Assumption 4 ensures existence of primal variable for the OCP, Assumption 3 guarantee the existence of dual variables that satisfy its KKT system. By applying (69) and replacing the dual variables of the KKT of the game with the OCP dual variables for every agent, we obtain a system of equations where the only unknowns are the user strategies. This system is similar to the OCP in the primal variables. Therefore, the OCP primal solution also satisfies the KKT necessary conditions of the game. Moreover, from the potentiality condition, it is straightforward to show that this primal solution of the OCP is also a PCL-NE of the MPG (see also (Zazo et al., 2016a, Theorem 1)).

Introduce the following vector field:

$$F(x_i, w, \boldsymbol{\sigma}_i) \triangleq \nabla_{(x_i, w)} J(x_i, \pi(x_i, w), \boldsymbol{\sigma}_i). \tag{70}$$

Since $F$ is conservative by construction (Apostol, 1969, Theorems 10.4, 10.5 and 10.9), conditions (67)–(68) are equivalent to (17)–(19) and we can calculate a potential $J$ through line integral (20). ∎

## D  PROOF OF THEOREM 2

*Proof:* We can rewrite game (8) by making explicit that the actions result from the policy mapping, which yields an expression that reminds the OL problem but with extra constraints:

$$
\mathcal{G}_3: \\
\forall k \in \mathcal{N}
\quad
\begin{aligned}
&\underset{w_k \in \mathbb{W}_k,\, \{\boldsymbol{a}_{k,i}\}_0^\infty \in \prod_{i=0}^\infty \mathbb{A}_k}{\text{maximize}} && \mathbb{E}\left[\sum_{i=0}^\infty \gamma^i r_k\left(\boldsymbol{x}_{k,i}^\tau, \boldsymbol{a}_{k,i}, \boldsymbol{a}_{-k,i}, \boldsymbol{\sigma}_{k,i}\right)\right] \\
&\text{s.t.} && \boldsymbol{a}_{k,i} = \pi(\boldsymbol{x}_{k,i}^\pi, w_k), \quad \boldsymbol{a}_{-k,i} = \pi(\boldsymbol{x}_{-k,i}^\pi, w_{-k}), \\
& && \boldsymbol{x}_{i+1} = f(\boldsymbol{x}_i, \boldsymbol{a}_i, \boldsymbol{\theta}_i), \\
& && g(\boldsymbol{x}_i, \boldsymbol{a}_i) \le 0,
\end{aligned}
\tag{71}
$$

where it is clear that: $\boldsymbol{a}_i \triangleq (\boldsymbol{a}_{k,i}, \boldsymbol{a}_{-k,i}) = \pi(\boldsymbol{x}_i, w)$ Rewrite also OCP (16) with explicit dependence on the actions:

$$
\mathcal{P}_2:
\quad
\begin{aligned}
&\underset{w \in \mathbb{W},\, \{\boldsymbol{a}_i\}_0^\infty \in \prod_{i=0}^\infty \mathbb{A}}{\text{maximize}} && \mathbb{E}\left[\sum_{i=0}^\infty \gamma^i J(\boldsymbol{x}_i, \boldsymbol{a}_i, \boldsymbol{\sigma}_i)\right] \\
&\text{s.t.} && \boldsymbol{a}_i = \pi(\boldsymbol{x}_i, w), \\
& && \boldsymbol{x}_{i+1} = f(\boldsymbol{x}_i, \boldsymbol{a}_i, \boldsymbol{\theta}_i), \\
& && g(\boldsymbol{x}_i, \boldsymbol{a}_i) \le 0.
\end{aligned}
\tag{72}
$$

By following the Euler-Lagrange approach described in Theorem 1, we have that the KKT systems for game and OCP are equal if the dual variables are equal (including new extra dual variables for the equality constraints that relate the action and the policy) and the following first-order conditions hold $\forall k \in \mathcal{N}$ and $i = 1, \ldots, \infty$:

$$\mathbb{E}\left[\nabla_{x_{k,i}^r} r_k\left(x_{k,i}^r, a_{k,i}, a_{-k,i}, \boldsymbol{\sigma}_{k,i}\right)\right] = \mathbb{E}\left[\nabla_{x_{k,i}^r} J\left(x_i, a_i, \boldsymbol{\sigma}_i\right)\right], \tag{73}$$

$$\mathbb{E}\left[\nabla_{a_{k,i}} r_k\left(x_{k,i}^r, a_{k,i}, a_{-k,i}, \boldsymbol{\sigma}_{k,i}\right)\right] = \mathbb{E}\left[\nabla_{a_{k,i}} J\left(x_i, a_i, \boldsymbol{\sigma}_i\right)\right]. \tag{74}$$

The benefit of this reformulation is that the gradient in (73) is taken with respect to the components in $\mathcal{X}_k^r$ only (instead of the whole set $\mathcal{X}$), at the cost of replacing (68) with the sequence of conditions (74). We have to realize that $a_{k,i}$ is indeed a function of variables $x_{k,i}^\pi$ and $w_k$. In order to understand the influence of this variable change, we use the identity $a_{k,i} = \pi_{w_k}(x_{k,i}^\pi)$ and apply the chain rule to both sides of (74), obtaining:

$$\mathbb{E}\left[\nabla_{x_{k,i}^\pi} r_k\right] = \mathbb{E}\left[\nabla_{x_{k,i}^r} r_k\right]^\top \nabla_{x_{k,i}^\pi} x_{k,i}^r + \mathbb{E}\left[\nabla_{a_{k,i}} r_k\right]^\top \nabla_{x_{k,i}^\pi} a_{k,i}, \tag{75}$$

$$\mathbb{E}\left[\nabla_{w_k} r_k\right] = \mathbb{E}\left[\nabla_{a_{k,i}} r_k\right]^\top \nabla_{w_k} a_{k,i}, \tag{76}$$

$$\mathbb{E}\left[\nabla_{x_{k,i}^\pi} J\right] = \mathbb{E}\left[\nabla_{x_{k,i}^r} J\right]^\top \nabla_{x_{k,i}^\pi} x_{k,i}^r + \mathbb{E}\left[\nabla_{a_{k,i}} J\right]^\top \nabla_{x_{k,i}^\pi} a_{k,i}, \tag{77}$$

$$\mathbb{E}\left[\nabla_{w_k} J\right] = \mathbb{E}\left[\nabla_{a_{k,i}} J\right]^\top \nabla_{w_k} a_{k,i}. \tag{78}$$

From (73)–(74), it is clear that the right side of (75) and (77) are equal. Similarly, from (74), the right side of (76) and (78) are equal, so that their left side must be also equal. Hence, we can replace (74) with the two following conditions:

$$\mathbb{E}\left[\nabla_{x_{k,i}^\pi} r_k\left(x_{k,i}^r, \pi_{w_k}\left(x_{k,i}^\pi\right), a_{-k,i}, \boldsymbol{\sigma}_{k,i}\right)\right] = \mathbb{E}\left[\nabla_{x_{k,i}^\pi} J\left(x_i, \pi_{w_k}\left(x_{k,i}^\pi\right), a_{-k,i}, \boldsymbol{\sigma}_i\right)\right], \tag{79}$$

$$\mathbb{E}\left[\nabla_{w_k} r_k\left(x_{k,i}^r, \pi_{w_k}\left(x_{k,i}^\pi\right), a_{-k,i}, \boldsymbol{\sigma}_{k,i}\right)\right] = \mathbb{E}\left[\nabla_{w_k} J\left(x_i, \pi_{w_k}\left(x_{k,i}^\pi\right), a_{-k,i}, \boldsymbol{\sigma}_i\right)\right]. \tag{80}$$

Moreover, we can combine (73) and (79) in one single equation:

$$\mathbb{E}\left[\nabla_{x_{k,i}^\ominus} r_k\left(x_{k,i}^r, \pi_{w_k}\left(x_{k,i}^\pi\right), a_{-k,i}, \boldsymbol{\sigma}_{k,i}\right)\right] = \mathbb{E}\left[\nabla_{x_{k,i}^\ominus} J\left(x_i, \pi_{w_k}\left(x_{k,i}^\pi\right), a_{-k,i}, \boldsymbol{\sigma}_i\right)\right]. \tag{81}$$

By using the identity $a_{-k,i} = \pi_{w_{-k}}(x_{-k,i}^\pi)$ in (80)–(81), we have:

$$\mathbb{E}\left[\nabla_{x_{k,i}^\ominus} r_k\left(x_{k,i}^r, \pi_{w_k}\left(x_{k,i}^\pi\right), \pi_{w_{-k}}\left(x_{-k,i}^\pi\right), \boldsymbol{\sigma}_{k,i}\right)\right] = \mathbb{E}\left[\nabla_{x_{k,i}^\ominus} J\left(x_i, \pi_u\left(x_i\right), \boldsymbol{\sigma}_i\right)\right], \tag{82}$$

$$\mathbb{E}\left[\nabla_{w_k} r_k\left(x_i, \pi_{w_k}\left(x_{k,i}^\pi\right), \pi_{w_{-k}}\left(x_{-k,i}^\pi\right), \boldsymbol{\sigma}_{k,i}\right)\right] = \mathbb{E}\left[\nabla_{w_k} J\left(x_i, \pi_u\left(x_i\right), \boldsymbol{\sigma}_i\right)\right]. \tag{83}$$

Note that under conditions (25)–(26), conditions (82)–(83) are equivalent to (67)–(68), with potential function $J$ equal to the objective of OCP (16). ∎

## E    PROOF OF PROPOSITION 1

*Proof:* Once that Theorem 2 has shown that the individual rewards can be expressed in separable form, it follows from the definition of proper function that: $r_k$ being proper implies that $J$ is also proper. Since $J$ is proper, it has nonempty level sets. Let $B \in \mathbb{R}$ define a nonempty level set of $J$:

$$\{a_0 \in \mathbb{C}_0, (x_i, a_i) \in \mathbb{C}_i : \mathbb{E}\left[J\left(x_i, a_i, \sigma_i\right)\right] \geq B\}_{i=0}^\infty. \tag{84}$$

Since $\gamma < 1$, we have:

$$\sum_{i=0}^\infty \gamma^i \mathbb{E}\left[J\left(x_i, a_i, \sigma_i\right)\right] \geq B \sum_{i=0}^\infty \gamma^i = \frac{B}{1-\gamma}. \tag{85}$$

Hence, the following level sets are also nonempty:

$$\left\{(x_i, a_i) : \sum_{i=0}^\infty \gamma^i \mathbb{E}\left[J\left(x_i, a_i, \sigma_i\right)\right] \geq \frac{B}{1-\gamma}\right\}_{i=0}^\infty. \tag{86}$$

In addition, since $J$ is proper, it must be upper bounded, i.e., $\exists U \in \mathbb{R}$, such that $J \leq U$. Then, we have:

$$\sum_{i=0}^{\infty} \gamma^i \mathbb{E}\left[J\left(x_i, a_i, \sigma_i\right)\right] \leq U \sum_{i=0}^{\infty} \gamma^i = \frac{U}{1-\gamma}. \tag{87}$$

Since $B \leq U$, we have that

$$\frac{B}{1-\gamma} \leq \frac{U}{1-\gamma}. \tag{88}$$

Therefore, the level sets (86) are bounded.

From Assumption 2 the fact that $J$ can be obtained from line integral (20), and fundamental theorem of calculus, we deduce that $J$ is continuous. Therefore, we conclude that these level sets are also compact. Thus, we can use (Bertsekas, 2007, Prop. 3.1.7, see also Sections 1.2 and 3.6) to ensure existence of an optimal policy. ∎

