# OpenReview forum: "Learning Parametric Closed-Loop Policies for Markov Potential Games"
_ICLR.cc/2018/Conference — Accept (Poster)_

### Official Review · AnonReviewer1 · 2017-11-27
**Interesting work on Markov potential games, from the viewpoint of someone without any prior knowledge on the topic**

**Rating:** 6
**Confidence:** 1

**Review:**

This manuscript considers a subclass of stochastic games named Markov potential games. It provides some assumptions that guarantee that a game is a Markov potential game and leads to some nice properties to solve the problem to approximately a Nash equilibrium. It is claimed that the work extends the state of the art by analysing the closed-loop version in a different manner, firstly constraining policies to a parametric family and then deriving conditions for that, instead of the other way around. As someone with no knowledge in the topic, I find the paper interesting to read, but I have not followed any proofs. The experimental setup is quite limited, even though I believe that the intention of the authors is to provide some theoretical ideas rather than applying them. Minor point: there are a few sentences with small errors, this could be improved.

---

> ### Author Response · Authors · 2017-12-28
> **We appreciate the feedback from the reviewer. We wish to emphasize the importance of providing a rigorous and effective method for finding closed-loop solutions to a relevant class of games. This application of learning representation advances the state of the art in multiagent systems.**
>
> We appreciate the feedback from the reviewer. We just wish to emphasize the importance of providing an analysis and effective method for finding closed-loop (CL) solutions for a relevant class of games that appear often in engineering and economics, and that includes cooperative and congestion games. Up to the best of our knowledge, this is the first time that this kind of solutions are rigorously provided for any class of Markov games with continuous variables and/or coupled constraints that appear often in engineering applications.
>
> Moreover, we remark that since our solution relies on parametric policies, being able to learn features is key for the applicability of the method. In summary, we believe this paper provides a useful application of representation learning for multiagent systems, which extends previous approaches, which only considered cooperative games or assumed finite state-action sets.
>
> We acknowledge that the experimental setup is limited. But as the reviewer suggests, our intention with the example in Appendixes A-B and with the numerical experiment in Sec. 5 is to illustrate how to apply the proposed framework to economic and engineering problems.

---

### Official Review · AnonReviewer2 · 2017-11-28
**ICLR may not be the right venue; Technical questions: Unclear how to deal with stochastic dynamics, etc.**

**Rating:** 6
**Confidence:** 3

**Review:**

Summary:
This paper studies multi-agent sequential decision making problems that belong to the class of games called Markov Potential Games (MPG). It considers finding the optimal policy within a parametric space of policies, which can be represented by a function approximator such as a DNN.
A main contribution of this work is that it shows that for MPG, instead of solving a multi-objective optimization problem (Eq. 8), which is difficult, it is sufficient to solve a scalar-valued optimization problem (Eq. 16).  Theorem 1 shows that under certain conditions on the reward function, the game is MPG. It also shows how one might find the potential function J, which is used in the single objective optimization problem.
Finding J can be computationally expensive in general. So the paper provides some properties that lead to finding J easier. For example, obtaining J is easy if we have a cooperative game (Corollary 1) or the reward can be decomposed/decoupled in a certain way (Theorem 2).


Evaluation:

This is a well-written paper that studies an important problem, but I don’t think ICLR is the right venue for it. There is not much about (representation) learning in this work. The use of TRPO as an RL algorithm in the Experiment does not play a critical role in this work either. Aside this general comment, I have several other more specific comments.


- There is a significant literature on the use of RL for multi-agent systems. The paper does not do a good job comparing and positioning with respect to them. For example, refer to the following recent paper and references therein:

Perolat, Strub, et al., “Learning Nash Equilibrium for General-Sum Markov Games from Batch Data,” AISTATS, 2017.


- If I understand correctly, the policies are considered to be functions from the state of the system to a continuous action. So it is a function, and not a probability distribution. This means that the space of considered policies correspond to the space of pure strategies. We know that for some games, the Nash equilibrium is a mixed strategy. Isn’t this a big limitation of this approach?


- I am unclear how this approach can handle stochastic dynamics. For example, the optimization (P1) depends on the realization of (theta_i)_i. But this is not available. The dependence is not only in the objective, but also in the constraints, which makes things more difficult.

I understand that in the experiments the authors used two models (either the average of random realization, or solving a different optimization for each realization), but none of them is an appropriate solution for a stochastic system.


- How large is the MPG class? Is there any structural result that positions them compared to other Markov Games? For example, is the class of zero-sum games an example of MPG?


- There is a comment close to the end of Section 5 that when there is no prior knowledge of the dynamics and the reward, one can use the proposed approach to learn PCL-NE by using any DRL.
This is questionable because if the reward is not known, the conditions of Theorems 1 or 2 cannot be verifies, so it is not possible to use (P1) instead of (G2).


- What comments can you make about the computational complexity? It seems that depending on the dynamics, the optimization problem P1 can be non-convex, hence computationally difficult to solve.


- How is the work related to the following paper?
Macua, Zazo, Zazo, “Learning in Constrained Stochastic Dynamic Potential Games,” ICASSP, 2016

======
I updated the score based on the authors' rebuttal.

---

> ### Author Response · Authors · 2017-12-28
> **We appreciate the careful reading and the detailed feedback. We believe to have addressed all concerns, including the motivation of the paper as a relevant application of learning representation. We will be glad to address any further concern.**
>
> We believe that ICLR is a propper venue. Our key contribution is to show that closed-loop NE (CL-NE) can be approximated with parametric policies. However, the applicability of this result is limited by the accuracy of the approximation. Approximations that depend on hand-coded features usually require domain knowledge and have to be re-designed for every game; while learned features that can express complex policies can alleviate these problems. Thus, we see this work as a relevant application of learned representations to multiagent systems that extend previous works, which only studied cooperative games, or assumed discrete state-action with no coupled constraints.
>
> Although the focus of our literature review is potential games, we know no previous method for approximating CL-NE for any class of Markov games with continuous variables and coupled constraints. There are open-loop (OL) analysis of some games with continuous variables, like monotone games. Also, (Perolat et al. 2017) studied state-dependent policies but assumed finite state-action sets, which are less common in engineering.
>
> Considering deterministic policies is not a limitation of our setting for two reasons: 1) Prop. 1 shows that under mild conditions, there exists a deterministic policy that achieves the optimal value of P1 and that is also an NE of G2. 2) We do not claim that our method will find all possible NE of G2, but just the one that is also solution to P1. There may be many (possible mixed strategies) solutions to G2, but we propose a method to find one of them.
>
> The reviewer has concerns about handling stochastic dynamics. We remark that the notation for objective and dynamix is standard in the literature. On the other hand, we agree that we should clearify that the optimal value of the OCP is the one that maximizes the expected return, for which the constraints are satisfied almost surely.
>
> Regarding the models used in the experiment, we remark that these two models are only for estimating the benchmark solution. The proposed DRL solution tackles the problem without taking into account any of these models. However, we are happy to change the way of computing the benchmark solution, and any further feedback on this direction will be much appreciated.
>
> The reviewer asks how large is the MPG class, and if zero sum games are an example of MPG. MPGs appear often in engineering and economics applications, where multiple agents have to share some resource. We have studied MPGs with "exact potentiality" condition, that includes cooperative and congestion games. There is a larger family of games that satisfy the "weighted potentiality" condition, where an agent’s change in reward due to its unilateral strategy deviation is equal to the change in the potential function but scaled by a positive weight. It is easy to show that weighted potential games (WPGs) and exact potential games can be made equivalent by scaling the reward functions [1, Lemma 2.1]. Thus, equivalent results to those presented here should be equally available for WPGs. A zero sum game is a WPG with weights 1 and -1, but we believe our KKT approach still holds in this case.
>
> The reviewer argues that it is not possible to learn PCL-NE with no prior knowledge of the environment, since Theorems 1 or 2 cannot be verified. We have to distinguish designer from DRL agents. Our claim is that we can use the proposed approach to find a PCL-NE by using any DRL agent that has no prior knowledge of the dynamics and/or the reward, given that the game is MPG. We do not claim that the agents are able to validate the Theorems. This situation is similar to previous works that assumed knowledge that the game is cooperative, or for most of the single agent reinforcement learning literature that assumes that the environment is an MDP without requiring the agents to verify it.
>
> The reviewer suggests that since the rewards are nonconvex, the computational complexity of P1 can be high. We disagree in part. Under Assumptions 1-4, having a discount factor smaller than one makes the Bellman operator monotone, independent on the convexity of the rewards. On the other hand, training a DRL algorithm implies finding local optima of nonconvex problems; but we remark that this is independent on the convexity of the agents' rewards.
>
> There are a number of notable differences with (Macua, Zazo, Zazo, 2016). The main one is that although such work had the intuition that MPGs could be solved with RL methods, it only included an OL analysis; actually, it only extended previous OL analysis to the stochastic case. That is the reason why it didn't consider state-dependent policy and their Corollary 1 missed the disjoint state condition. Since such OL analysis is not satisfactory for stochastic dynamics, the current paper bridges this gap. We believe that this is an important piece in the potential games literature.
>
> [1] Lã et al. Potential game theory: applications in radio resource allocation. Springer, 2016

---

### Official Review · AnonReviewer4 · 2017-12-19

**Rating:** 7
**Confidence:** 2

**Review:**

While it is not very surprising that in a potential game it is easy to find Nash equilibria (compare to normal form static games, in which local maxima of the potential are pure Nash equilibria), the idea of approaching these stochastic games from this direction is novel and potentially (no pun intended) fruitful. The paper is well written, the motivation is clear, and some of the ideas are non-trivial. However, the connection to learning representations is a little tenuous.

---

> ### Author Response · Authors · 2017-12-28
> **We thank the reviewer for the good feedback. Since our key idea is to rely on expressive parametric policies, we believe this work presents a relevant application of learning representation for multiagent systems.**
>
> We also expected that finding closed-loop Nash equilibria in MPG should be doable. However, we remark that the closed-loop analysis is much more slippery than the open-loop analysis, since the agents have to take into account not only all possible trajectories over the state-action space (as in the open-loop case), but also all possible deviations from that trajectories at every step. The situation is even more involved since we consider coupled constraints (i.e., we are considering the stochastic infinite-horizon extension of a relevant class of generalized Nash equilibrium problems like those studied in [1]). Up to the best of our knowledge this is the first work that provides a rigorous analysis and an effective method for learning approximate closed-loop Nash equilibrium in  continuous MPG (actually in any class of games with continuous state-action variables).
>
> The reviewer comments that the connection of the current work with learning representations is a little tenuous. Although the main focus of the paper is the theoretical analysis of Markov potential games (MPGs), we believe that this connection is indeed stronger than it might seem. Our key idea is to rely on parametric policies, whose applicability for real problems depends on the expressiveness of the parametric family. If the optimal policy is a complicated mapping from states to actions, we require sophisticated parametric approximations that are able to approximate such mapping. Parametric approximations that depend on hand-coded features usually require expert domain knowledge, can be time consuming (especially for multiagent problems), and have to be re-designed for every problem at hand; while learned features that can express complex closed-loop policies are able to alleviate these problems, hence, crucial to the usefulness of our method. In summary (as responded to AnonReviewer2), we see the current setting as a relevant application of learned representations that extend previous multiagent applications, which only studied cooperative games, or assumed discrete state-action, and never with coupled constraints. In addition, we remark that our analysis allows to reformulate the game in a centralized manner that could inspire the extension of advanced DRL techniques like [2, 3], which were previously only valid for cooperative games.
>
> [1] F. Facchinei and C. Kanzow. "Generalized Nash equilibrium problems." 4OR: A Quarterly Journal of Operations Research 5.3 (2007): 173-210.
>
> [2] J. Foerster et al. "Counterfactual Multi-Agent Policy Gradients." arXiv preprint arXiv:1705.08926 (2017).
>
> [3] P. Sunehag et al. "Value-Decomposition Networks For Cooperative Multi-Agent Learning." arXiv preprint arXiv:1706.05296 (2017).

---

### Author Response · Authors · 2018-01-04
**New version**

We have addressed the comments from the reviewers. In addition, we have strengthened the form of Theorem 2.

---

### Decision · Program_Chairs · 2018-01-29
**ICLR 2018 Conference Acceptance Decision**

**Decision:**

Accept (Poster)

**Comment:**

The paper considers Markov potential games (MPGs),  where the agents share some common resource. They consider MPGs with continuous state-action variables, coupled constraints and nonconvex rewards, which is novel. The reviews are all positive and point out the novel contributions in the paper